# The Toxicity of Salicylhydroxamic Acid and Its Effect on the Sensitivity of *Ustilaginoidea virens* to Azoxystrobin and Pyraclostrobin

**DOI:** 10.3390/jof8111231

**Published:** 2022-11-21

**Authors:** Jiehui Song, Zhiying Wang, Sijie Zhang, Yan Wang, You Liang, Qigen Dai, Zhongyang Huo, Ke Xu

**Affiliations:** Jiangsu Key Laboratory of Crop Genetics and Physiology & Co-Innovation Center for Modern Production Technology of Grain Crops, Agricultural College, Yangzhou University, Yangzhou 225009, China

**Keywords:** artificial media, fungicide, salicylhydroxamic acid (SHAM), toxicity, *Ustilaginoidea virens*

## Abstract

Rice false smut (RFS) caused by *Ustilaginoidea virens* has been one of the most severe rice diseases. Fungicide-based chemical control is a significant measure to control RFS. In the sensitivity determination of quinone outside inhibitor (QoI) fungicide in vitro, salicylhydroxamic acid (SHAM) has been commonly added to artificial culture media in order to inhibit alternative oxidase of phytopathogenic fungi. However, some studies showed that artificial media should not include SHAM due to its toxicity. Whether SHAM should be added in the assay of *U. virens* sensitivity to QoI fungicide remains unknown. In this study, two appropriate media, potato sucrose agar (PSA) and minimal medium (MM), were selected to test SHAM toxicity and sensitivity of *U. virens* to azoxystrobin and pyraclostrobin. The mycelial growth and sensitivity to azoxystrobin and pyraclostrobin had no significant difference between on PSA and MM. SHAM could significantly inhibit mycelial growth, conidial germination, peroxidase (POD) and esterase activity of *U. virens*. Average effective concentration for inhibiting 50% (EC_50_) values of SHAM against mycelial growth of ten *U. virens* were 27.41 and 12.75 μg/mL on PSA and MM, respectively. The EC_50_ values of SHAM against conidial germination of isolates HWD and JS60 were 70.36 and 44.69 μg/mL, respectively. SHAM at 30 μg/mL significantly inhibited POD and esterase activity of isolates HWD and JS60, and even SHAM at 10 μg/mL significantly inhibited POD activity of isolate HWD. In addition, SHAM significantly reduced EC_50_ values and EC_90_ values of azoxystrobin and pyraclostrobin on both PSA and MM. Even in the presence of SHAM at 10 μg/mL, average EC_50_ values of ten *U. virens* isolates for azoxystrobin decreased 1.7-fold on PSA and 4.8-fold on MM, and for pyraclostrobin that decreased 2.8-fold on PSA and 4.8-fold on MM. Therefore, these results suggest that SHAM should not be included in artificial media in the assay of *U. virens* sensitivity to QoI fungicides.

## 1. Introduction

Rice false smut (RFS) results from the filamentous fungus *Ustilaginoidea virens* (teleomorph: *Villosiclava virens*) and is responsible for significant losses in the rice industry. RFS has recently developed into one of the most severe rice diseases in most rice-producing areas worldwide [1]. The RFS pathogen (*U. virens*) infects the plant during the flowering stage [2,3,4]. The infected spikelets convert first into whitish, yellowish orange to green chlamydospores, which later turn black in colour [5]. Worse still, *U. virens* can also produce cyclopeptide mycotoxins. These cyclopeptide mycotoxins are toxic to humans and animals and threaten rice production and food security [6,7]. Although planting resistant varieties is undoubtedly an economical and effective method of controlling RFS, the majority of rice varieties with good quality and high yields are RFS-susceptible. So far, fungicide-based chemical control is a significant measure for controlling RFS [8,9].

With respect to market share and sales, quinone outside inhibitor (QoI) fungicides are the most significant class of agricultural fungicides. In particular, pyraclostrobin and azoxystrobin have been the most widely used. Due to their broad-spectrum activity, QoI fungicides have been extensively applied to a wide variety of crops, e.g., cucurbits, tomatoes, cereals, potatoes, pome fruits and oilseed rapes [10,11,12]. QoI fungicides have the following biochemical action mechanism: The electron transport is blocked at the quinol-oxidizing site of the cytochrome bc1 complex (complex III) in the mitochondrial respiration chain, and then the synthesis of adenosine triphosphate is inhibited [10]. QoI fungicides can effectively inhibit the mycelial growth, sporulation and spore germination of many fungal plant pathogens. In our previous study, azoxystrobin and pyraclostrobin had significant controlling effects on RFS in the crop field [13]. In 2014, azoxystrobin was officially registered in China to control *U. virens* (China Pesticide Information Network, http://www.chinapesticide.org.cn last accessed on 21 October 2022). In contrast, pyraclostrobin has not been registered to control *U. virens* by 2022. Currently, in China, no reports indicate that *U. virens* has resistance to QoI fungicides.

As a distinctive inhibitor of alternative oxidase (AOX), salicylhydroxamic acid (SHAM) has been commonly added to artificial culture media in order to inhibit AOX for the in vitro assay of the sensitivity to QoI fungicides [11,14,15,16]. Researchers have found that fungi could partly escape the in vitro toxicity of Qol fungicides by activating AOX located on mitochondrial membranes. The AOX is believed to be suppressed by plant secondary metabolites (e.g., flavonoids) [17,18]. It has been assumed that including SHAM in artificial media can allow better consistency between the in vitro bioassay to be and what occurs in planta. However, some studies showed that artificial media should not include SHAM in the assays of *Botrytis cinerea*, *Fusicladium effusum* and *Sclerotinia sclerotiorum* sensitivity to the QoI fungicides [19,20,21,22]. In fact, SHAM has complex and even unpredictable effects on the EC_50_ of some QoI fungicides in some cases. Azoxystrobin with SHAM against *Microdochium nivale* and *F. effusum* had higher EC_50_ values than azoxystrobin alone [22,23]. Under 100 μg/mL SHAM, azoxystrobin against *S. sclerotiorum* showed a lower EC_50_, while dicarboximide fungicide procymidone showed increasing EC_50_ [24]. The versatile activities of SHAM against plant pathogens have been another issue of concern. AOX regulates mycelial growth, development, and resistance to oxidative stress [25]. Apart from inhibiting AOX, SHAM also suppresses essential fungal enzymes, e.g., esterase, peroxidase, putative melanogenesis enzymes, and laccase [21,26,27]. The toxicity of SHAM against other fungal pathogens is also revealed. For instance, at a low concentration, SHAM could inhibit the mycelial growth of *F. effusum* [22]. Therefore, in the assay of *U. virens* sensitivity to QoI fungicides, whether SHAM should be added remains unknown.

Current protocols for fungicide sensitivity testing are mostly based on mycelial growth inhibition and spore germination inhibition. For mycelial growth inhibition, PSA is commonly used to determine the sensitivity of *U. virens* to fungicides [8,13]. However, different culture media can result in different sensitivities. For example, the sensitivity of *Monilinia fructicola* to boscalid showed great differences on potato dextrose agar (PDA) and minimal medium (MM) [28]. In addition, the sensitivity of *Ceratocystis fagacearums* to the triazoles was up to tenfold higher in neopeptone broth culture than on PDA [29]. Therefore, the selection of appropriate culture medium is very important for the determination of fungicide sensitivity and SHAM toxicity.

The objectives of this study were: (1) Select an appropriate medium for testing the sensitivity of *U. virens* to azoxystrobin and pyraclostrobin and SHAM toxicity. (2) Investigate the toxic effect of SHAM alone on *U. virens*. (3) Assess the effect of SHAM on the EC_50_ values of the two QoI fungicides azoxystrobin and pyraclostrobin.

## 2. Materials and Methods

### 2.1. U. virens Isolates

In this study, ten isolates of *U. virens* were taken from rice fields in different counties and cities in China. Individual RFS balls were used to obtain single isolates. These RFS balls were surface-sterilized for 5 min using 0.1% sodium hypochlorite, rinsed for 30 s using distilled sterile water, and then cut into several small pieces. The pieces were moved to potato sucrose agar (PSA) amended with streptomycin sulfate and incubated in a growth chamber at 27 °C for 3–7 d. Pure cultures were acquired by transferring individual germinated chlamydospore, and cultured on PSA at 27 °C for 14 days.

### 2.2. Fungicides and Media

In this study, technical-grade pyraclostrobin (97% active ingredient [a.i.], Jiangsu Gengyun Chemical Co. Ltd., Changzhou, Jiangsu, China) and azoxystrobin (96% a.i., Jiangsu Gengyun Chemical Co., Ltd., Jiangsu, China) were used. The fungicides were dissolved in acetone at a concentration of 10,000 µg a.i./mL to obtain stock solutions. The stock solutions of each fungicide were placed in a refrigerator at 4 °C for no more than two weeks prior to serial dilution for bioassay experiments.

A total of 200 g of potato and 20 g of sucrose per liter was used to prepare potato sucrose broth (PSB). 15 g of agar was added to 1 L of PSB to produce PSA. 1 L of minimal medium (MM) consisted of 2 g KH_2_PO_4_, 0.5 g MgSO_4_·7H_2_O, 1.5 g K_2_HPO_4_, 1 g (NH4)_2_SO_4_, 10 g glucose, 12.5 g agar, and 2 g yeast extract according to Hu et al. [28].

### 2.3. Determination of Sensitivity of U. virens to Azoxystrobin and Pyraclostrobin

Ten *U. virens* isolates were used to measure the EC_50_ (the concentration of fungicides resulting in 50% reduction in growth compared to unamended control) of azoxystrobin and pyraclostrobin against mycelial growth based on previous studies [13,30]. Azoxystrobin and pyraclostrobin were used to amend autoclaved PSA or MM. The obtained final concentrations of fungicides included 0, 0.01, 0.03, 0.1, 0.3, 1.0, 3.0, and 10.0 μg/mL. An inverted mycelial plug with a diameter of 4 mm was sampled at the edge of a 14-day-old colony and placed on 9 cm petri dishes with the amended medium. These plates were incubated in the dark for 21 days at 27 °C. Average colony diameter (minus the inoculation plug diameter) in each treatment was determined and represented by the growth inhibition percentage. The fungicide-free treatment was taken as control. Each experiment of isolate was performed in triplicate. Growth inhibition percentage (%) calculates with the following formula: (diameter of the control − diameter of the treatment)/diameter of the control × 100%. Regression analysis of corresponding growth inhibition percentage against the logarithm of fungicide concentrations was used to obtain EC_50_.

### 2.4. Toxicity of SHAM against Mycelial Growth of U. virens on PSA and MM

Technical-grade product of SHAM (99% a.i.; Syngenta Biotechnology Co. Ltd., Shanghai, China) was dispersed in methanol at the concentration of 20,000 µg a.i./mL. SHAM was added to obtain final concentrations of 0, 10, 20, 40, 80, and 160 µg/mL in PSA or MM. PSA or MM amended with only solvent was taken as the control. Then, mycelial plugs sampled from the periphery of 14-day-old colonies of ten isolates were placed on SHAM-amended PSA or MM plates. These plates were incubated at 27 °C for 21 d in the dark. The diameter of each colony was measured twice at right angles. For each treatment, the average colony diameter (minus the inoculation plug diameter) was determined and represented by the growth inhibition percentage. The EC_50_ values of SHAM were obtained using regression analysis as described in the previous section.

### 2.5. Toxicity of SHAM against Conidial Germination of U. virens

Two isolates (i.e., HWD and JS60) were used to investigate the toxicity of SHAM against conidial germination. Two mycelia plugs with a diameter of 4 mm were transferred into 50 mL of PSB and stirred at 27 °C, 150 rpm for 7 days. The hyphae were removed by filtration through six layers of gauze. The conidia were taken from the filtrate under centrifugation at 10,000 rpm for 5 min. The conidia were rinsed twice through resuspension in distilled sterile water. Then, they were resuspended and adjusted to reach a concentration of 1.0 × 10^5^ conidia/mL. SHAM was added to obtain final concentrations in water agar (WA: 15 g of agar per liter), i.e., 0, 10, 20, 40, 80, and 160 µg/mL. 100 μL conidial suspension was spread on the surface of WA plates amended with SHAM of the above concentrations. Then, these plates were incubated at 27 °C in dark for 24 h. The conidial germination rate was recorded with a Nikon E200 microscope. When the germ tube was longer than half of the conidia, the conidia was regarded as germinated. Germination was determined by counting not fewer than 200 conidia per plate for each isolate. Each isolate was repeated three times and the experiment was conducted twice.

### 2.6. Effects of SHAM on the Activity of Peroxidase (POD) and Esterase

The isolates (JS60 and HWD) grew on PSA plates covered with cellophane. PSA amended with SHAM at a final concentration of 0, 10 and 30 µg/mL was used to culture *U. virens*. When the concentration of SHAM was more than 30 µg/mL, the mycelium grew very little, which made it difficult to collect enough mycelium. Therefore, the concentrations of SHAM were set at 10 and 30 µg/mL, one of which was lower than EC_50_ of SHAM and the other of which was higher than EC_50_ of SHAM. PSA amended with the solvent alone was used as the control. Mycelia were obtained from 14-day-old colonies of the two isolates. Mycelial homogenization was conducted according to the methods of Duan et al. (2013) [31]. A total of 1 mL cold extraction buffer was used to homogenize about 0.3 g fresh mycelia of each isolate in order to determine the POD activity. The buffer contained 1% (*w*/*v*) polyvinylpolypyrrolidone (PVPP) and 50 mM sodium phosphate buffer (SPB) (pH 5.5). A total of 50 mM cold SPB (pH 5.5) without PVPP was used to homogenize mycelia to determine the esterase activity. The homogenate was centrifuged at 10,000× *g* at 4 °C for 20 min. The enzyme activity was analyzed using the supernatant.

POD activity was assayed based on previous studies with some modifications [21,31]. A total of 0.475 mL of 0.2% substrate guaiacol in 50 mM SPB and 40 μL of the supernatant were added to each reaction vessel. After the addition of 0.5 mL of 0.3% H_2_O_2_, the increase in absorbance at 470 nm was measured using spectrophotometry. A unit of enzyme activity (1 U) was characterized as one unit change of absorbance per minute.

Esterase activity was assessed using spectrophotometry based on previous studies [21,32]. The substrate chosen was p-nitrophenyl palmitate (pNPP). The reaction mixture contained 0.35 mL of 0.067 mmol/L magnesium chloride, 1 mL of 0.05 mol/L citrate buffer (pH 5.0), 50 µL of enzyme supernatant and 0.1 mL of 0.67 mg/mL pNPP. The mixture was incubated at 37 °C for 30 min. Then, the reaction was stopped by adding 0.2 mL of 2 mol/L sodium hydroxide to the mixture. Based on the absorbance at 405 nm, the reaction product (pnitrophenol (pNP)) was measured. An enzyme activity unit (1 U) was characterized as 1 µmol/min pNP formed.

### 2.7. Effects of SHAM on Sensitivity of U. virens to Azoxystrobin and Pyraclostrobin

To test the potentiation by SHAM of various concentrations, individual 4 mm diameter mycelium plugs were removed from the edge of a 14-day-old colony (isolates JS60 and HWD) on PSA and placed in the center of a 9 cm Petri dish. The dish contained PSA or MM amended with azoxystrobin or pyraclostrobin in the presence of SHAM at different concentrations. At 0 or 10 µg/mL of SHAM, the final tested azoxystrobin or pyraclostrobin concentrations were 0.01, 0.03, 0.1, 0.3, 1.0, or 3.0 µg/mL. At 20, 40, or 80 µg/mL of SHAM, the tested azoxystrobin or pyraclostrobin concentrations were 0.003, 0.01, 0.03, 0.1, 0.3, or 1.0 µg/mL. Then, mycelial growth was measured, and EC_50_ was calculated as described above. Each treatment was performed in triplicate. The experiment was repeated twice.

Considering that the SHAM with lower concentration has lower toxicity against mycelial growth, ten *U. virens* isolates were used to determine the changes in EC_50_ in the presence of SHAM at 10 µg/mL. To obtain EC_50_ under SHAM, PSA or MM at 55 °C were amended with azoxystrobin or pyraclostrobin at final concentrations of 0.01, 0.03, 0.1, 0.3, 1.0, or 3.0 µg/mL. SHAM was added to a duplicate set of fungicide-amended medium in order to obtain a final concentration of 10 µg/mL. Mycelial plugs with a diameter of 4 mm were obtained from the periphery of 14-day-old colonies and placed upside down in the center of the amended medium plates. The fungicide-free medium was amended with organic solvents and taken as the control. The experiments were conducted in triplicate and repeated twice.

### 2.8. Statistical Analysis

The significant differences of the POD and esterase activity among SHAM treatments with different concentrations were evaluated by SPSS software (version 22.0; IBM SPSS Inc. Chicago, IL, USA), using one-way analysis of variance (ANOVA) with a least significant difference (LSD) test. *T* test was used to compare the differences in mycelial growth rate and fungicide sensitivity between on PSA and MM. Before that, F-test was used to check whether the variance was homogeneous between the two groups. A paired *t* test was used to compare the difference between EC_50_ with and without SHAM.

## 3. Results

### 3.1. Sensitivity of U. virens Isolates to Azoxystrobin and Pyraclostrobin

The average mycelial growth rate of ten isolates of *U. virens* on PSA was 2.11 mm/d, and that on MM was 1.96 mm/d (Table 1). Statistical analysis showed that there were no significant differences (*p* = 0.430) in mycelial growth rate between isolates grown on PSA and MM. As for the sensitivity to azoxystrobin, the average EC_50_ of ten isolates on PSA was 0.156 μg/mL, and that on MM was 0.135 μg/mL. For pyraclostrobin, the average EC_50_ on PSA and MM were 0.070 and 0.077 μg/mL, respectively. There were no significant differences in average EC_50_ between PSA and MM, with *p* = 0.610 for azoxystrobin and *p* = 0.615 for pyraclostrobin.

### 3.2. Toxicity of SHAM against Mycelial Growth of U. virens on PSA and MM

As shown in Figure 1, SHAM showed strong toxicity to *U. virens* isolate HWD on both PSA and MM media. The average inhibition percentage of mycelial growth of ten isolates increased linearly with the multiple increases in the concentration of SHAM (Figure 2). On both PSA and MM amended with SHAM at 10 to 160 μg/mL, all 10 isolates tested showed significant growth inhibition (*p* < 0.05). When the concentration of SHAM was 10 μg/mL, the average inhibition percentage of the 10 isolates on PSA was 28.0 ± 12.5% (standard deviation), and that on MM was 47.3 ± 14.4%. When the concentration of SHAM was 160 μg/mL, the average inhibition percentage of the 10 isolates on PSA was 99.8 ± 0.5%, and that on MM was 100.0 ± 0%.

The ranges of EC_50_ for SHAM were from 7.85 to 41.68 μg/mL on PSA and 4.47 to 20.73 μg/mL on MM, with means of 27.41 and 12.75 μg/mL on PSA and MM, respectively (Table 2). The fold change (PSA vs. MM) of EC_50_ of SHAM ranged from 1.56 to 5.20, with a mean of 2.30. Statistical analysis showed that different media had a significant effect (*p* = 0.003) on the EC_50_ of SHAM. These results indicated that the toxicity of SHAM to *U. virens* on MM was significantly stronger than that on PSA medium.

### 3.3. Toxicity of SHAM against Conidial Germination of U. virens

SHAM at concentrations ranging from 20 to 160 μg/mL had significant inhibitions (*p* < 0.05) on the conidial germination of *U. virens* isolates HWD and JS60 (Figure 3). Compared with the toxicity against mycelial growth, SHAM had lower inhibition effects on conidial germination of *U. virens* isolates. The EC_50_ values of SHAM against conidial germination of isolates HWD and JS60 were 70.36 and 44.69 μg/mL, respectively.

### 3.4. Effects of SHAM on Activity of POD and Esterase

POD activity decreased significantly (*p* < 0.05) for mycelia of isolate HWD grown on PSA supplemented with 10 and 30 μg/mL SHAM, by 16.7% and 66.7%, respectively. For isolate JS60, no significant effect on POD activity was detected at 10 μg/mL SHAM, but POD activity decreased significantly (*p* < 0.05) by 24.5% at 30 μg/mL SHAM (Figure 4A).

Esterase activity of two *U. virens* isolates was not significantly affected at 10 μg/mL SHAM. However, esterase activity was significantly inhibited (*p* < 0.05) by 39.7% for isolate HWD and 29.9% for isolate JS60 at 30 μg/mL SHAM (Figure 4B).

### 3.5. Effects of SHAM on Sensitivity of U. virens to Azoxystrobin and Pyraclostrobin

The inhibition of mycelial growth of *U. virens* by azoxystrobin or pyraclostrobin with SHAM was greater than that without SHAM on both PSA and MM. With the increase of SHAM concentration, the sensitivity of *U. virens* to azoxystrobin and pyraclostrobin increased (Table 3).

In the presence of SHAM at 10 μg/mL, average EC_50_ for the ten *U. virens* isolates for azoxystrobin and pyraclostrobin were considerably lower on both PSA and MM (Figure 5). Of the ten tested isolates, individual EC_50_ values of only two isolates for azoxystrobin on PSA amended with 10 μg/mL SHAM were higher than those without SHAM (Appendix A). The average EC_50_ of the ten isolates for azoxystrobin on PSA with SHAM was 0.093 μg/mL, whereas that was 0.155 μg/mL without SHAM. On MM, the average EC_50_ was 0.028 μg/mL with SHAM, whereas that without SHAM was 0.135 μg/mL. The fold change (without SHAM vs. with SHAM) of EC_50_ for azoxystrobin on PSA ranged from 0.82 to 3.70, with a mean of 1.67, and that on MM ranged from 1.48 to 16.97, with a mean of 4.82. Based on results of a paired *t* test of the EC_50_, there was a significant difference with *p* = 0.020 for PSA and *p* < 0.001 for MM between average EC_50_ of azoxystrobin with and without SHAM (Figure 5A). As for pyraclostrobin, all the ten isolates tested were more sensitive on both PSA and MM amended with SHAM, compared with those without SHAM (Appendix A). The average EC_50_ of pyraclostrobin were 0.025 and 0.016 μg/mL on PSA and MM amended with SHAM, respectively, which were much lower than those without SHAM. The fold change (without SHAM vs. with SHAM) of EC_50_ values for pyraclostrobin on PSA ranged from 1.38 to 11.86, with a mean of 2.80, and that on MM ranged from 1.09 to 12.57, with a mean of 4.81. Similar to azoxystrobin, there was a significant difference with *p* < 0.001 for both PSA and MM between average EC_50_ values of pyraclostrobin with and without SHAM (Figure 5B).

## 4. Discussion

Fungicide sensitivity tests using mycelial growth inhibition may not be as fast as testing conidial germination and growth or germination rates, but one of the most important advantages is that results can be assessed visually without the need for a spectrometer or microscope. PSA has been used to determine the sensitivity of *U. virens* to sterol demethylation inhibitor (DMI) fungicides and QoI fungicides by mycelial growth inhibition [8,13,33]. MM has been used for different fungi to test succinate dehydrogenase inhibitor (SDHI) fungicide sensitivity [28,34,35]. For *U. virens*, MM used as a culture medium has not been reported. In this study, MM supported the mycelial growth of *U. virens* as well as PSA, and the EC_50_ values of azoxystrobin and pyraclostrobin for *U. virens* had no significant difference between MM and PSA. Additionally, MM yielded lower EC_50_ values of SHAM than PSA, which could better reflect the inherent toxicity of SHAM. Therefore, MM is an appropriate medium for determining the sensitivity of *U. virens* to azoxystrobin and pyraclostrobin and SHAM toxicity.

Incorporating SHAM into artificial media is a well-established practice for in vitro assays of fungal pathogen sensitivity to QoI fungicides [11,14,15,23]. SHAM at a final concentration of 100 μg/mL or 0.5 mM is typically added to an artificial medium (0.653 mM = 100 μg/mL). SHAM toxicity has not been reported in previous in vitro studies of several other fungi, including *Phoma ligulicola* (0.653 mM SHAM), *Alternaria solani* (0.653 mM SHAM), *Botrytis cinerea* (0.5 mM SHAM), *Cercospora beticola* (0.5 mM SHAM) and *Ustilago maydis* (0.5 mM SHAM) [14,36,37,38,39], but other reports revealed that SHAM had limited toxic effects. In *Colletotrichum graminicola*, fungal inhibition by SHAM (0.653 mM) did not exceed 30% [40]. In *Magnaporthe oryzae*, 0.653 mM SHAM inhibited conidial germination by 3–14% [41]. However, some studies showed that SHAM had strong toxicity to fungal mycelial growth. SHAM at 100 μg/mL showed significant inhibitory activity against the mycelial growth of *Mycosphaerella citri, Elsinoe fawcettii* and *Diaporthe citri* [42]. Studies on *F. effusum* demonstrated that the eight isolates exhibited inhibitory effects on the growth by over 50% in potato dextrose broth media amended with SHAM at 0.5 or 1.0 mM [22]. In addition, two studies found that the average EC_50_ value of SHAM against the mycelial growth of *S. sclerotiorum* was 97.5 μg/mL and 54.8 μg/mL, respectively [20,21]. In the present study, the average EC_50_ values of SHAM against mycelial growth of *U. virens* were 27.41 and 12.75 μg/mL on PSA and MM, respectively, far below 100 μg/mL. The EC_50_ values of SHAM against conidia germination of two isolates were also lower than 100 μg/mL. Thus, it can be seen that SHAM has high toxicity to various phytopathogenic fungi. This indicates that regularly adding SHAM to artificial media is not always suitable when assaying fungal sensitivity to QoI fungicides.

In the present study, SHAM significantly decreased the EC_50_ of azoxystrobin and pyraclostrobin, even at the concentration of 10 μg/mL. With the increase in the SHAM concentration, the EC_50_ or EC_90_ of azoxystrobin and pyraclostrobin decreased. The combination of SHAM and azoxystrobin or pyraclostrobin was much more inhibitory than azoxystrobin or pyraclostrobin alone. The results were similar to those for *B. cinerea* and *S. sclerotiorum*: adding SHAM decreased the EC_50_ of QoI fungicides [11,19,21]. SHAM increased the inhibition, indicating its effects on alternative respiration. However, previous studies showed that SHAM significantly improved the control efficacy of dicarboximide fungicide dimethachlone and QoI fungicides against *B. cinerea* and *S. sclerotiorum*. The increase in the control efficacy of QoI fungicides was lower than that of dimethachlone [21]. Theoretically, the control efficacy of QoI fungicides would not be significantly affected by SHAM if SHAM has the only function of inhibiting AOX. AOX was inhibited by secondary metabolites of plants. Furthermore, the strong synergistic effect of SHAM on the control efficacy of dimethachlone is difficult to explain since mitogen-activated protein/histidine kinase in osmotic signal transduction is the action target of dimethachlone. These results indicated that SHAM may not only inhibit the alternative respiration of fungi but also has potential cytotoxicity.

It is worth noting that SHAM does not always reduce the EC_50_ of QoI fungicides. In the presence of 0.5 mM SHAM, EC_50_ of azoxystrobin against *Microdochium nivale* showed an increase compared with that without SHAM [23]. The average EC_50_ of azoxystrobin with 100 μg/mL SHAM against 18 isolates of *F. effusum* was over twofold higher than that without SHAM [22]. EC_50_ of kresoxim-methyl with 100 μg/mL SHAM against the conidia germination of *Cladosporium caryigenum* was over fivefold higher than that without SHAM [43]. Xu et al. (2013) reported that compared with that without SHAM, the EC_50_ of azoxystrobin with 100 μg/mL SHAM against *S. sclerotiorum* significantly decreased, while the dicarboximide fungicide procymidone with SHAM significantly showed an increased EC_50_ [24]. These findings indicate that based on certain combinations of fungal pathogens and fungicides, EC_50_ of fungicides may be significantly increased or reduced by SHAM.

SHAM has been found to inhibit some essential fungal enzymes, e.g., putative melanogenesis enzymes, POD, laccase, and esterase [21,26,27,44,45]. The present study demonstrated that SHAM at 30 μg/mL significantly inhibited POD and esterase activity of two *U. virens* isolates. Even at 10 μg/mL, SHAM significantly inhibited POD activity of *U. virens* isolates HWD. A previous study showed that SHAM had significant inhibitory effects on esterase of *S. sclerotiorum* and POD of *B. cinerea* [21]. In addition, it is shown that AOX can regulate mycelial growth and development of fungi [25]. To know the function of AOX in *U. virens*, the AOX protein of *U. virens* (XP_042998005.1, 352 amino acids) was compared with that of other fungi. The homology comparison by BlastP showed that AOX of *U. virens* has up to 80.95% identity to that of *Metarhizium anisopliae* (KJK83722.1), 80.67% identity to that of *Pochonia chlamydosporia* (XP_018141496.1) and *Metarhizium acridum* (XP_007810939.1), 80.11% identity to that of *Metarhizium robertsii* (EXV01183.1), and 80.06% identity to that of *Claviceps* aff. *purpurea* (KAG6294413.1). The specific function of AOX in *U. virens* needs further study. The inhibitory effects of SHAM on AOX and other essential enzymes and the multiple functions of AOX imply that EC_50_ values of fungicides may be significantly affected by SHAM, thus inducing erroneous bioassay results.

## 5. Conclusions

In conclusion, this study demonstrated that the mycelial growth and sensitivity of *U. virens* to azoxystrobin and pyraclostrobin had no significant difference between on PSA and MM. SHAM could significantly inhibit mycelial growth and conidial germination of *U. virens*. SHAM at 30 μg/mL significantly inhibited POD and esterase activity of two *U. virens* isolates. SHAM significantly reduced the EC_50_ values of azoxystrobin and pyraclostrobin on both PSA and MM, even at 10 μg/mL SHAM. These results indicate that the bioassay results obtained in the presence of SHAM cannot reflect the intrinsic sensitivity of pathogens to the fungicides tested due to evidence of in vitro toxicity of SHAM. Therefore, SHAM should not be included in artificial media in the assay of *U. virens* sensitivity to QoI fungicides.

## Figures and Tables

**Figure 1 jof-08-01231-f001:**
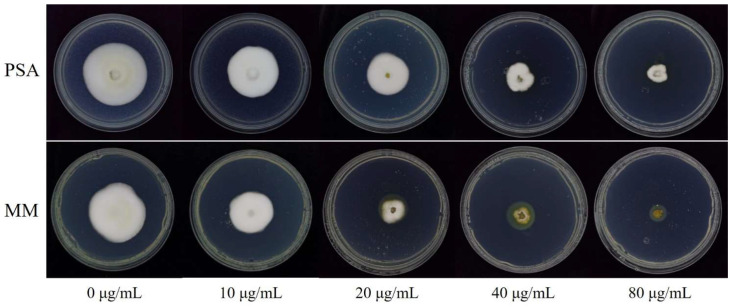
Effects of SHAM at different concentrations on mycelial growth of *Ustilaginoidea virens* isolate HWD on PSA and MM. Photos were taken after 21 days of incubation at 27 °C in the dark. PSA, potato sucrose agar; MM, minimal medium.

**Figure 2 jof-08-01231-f002:**
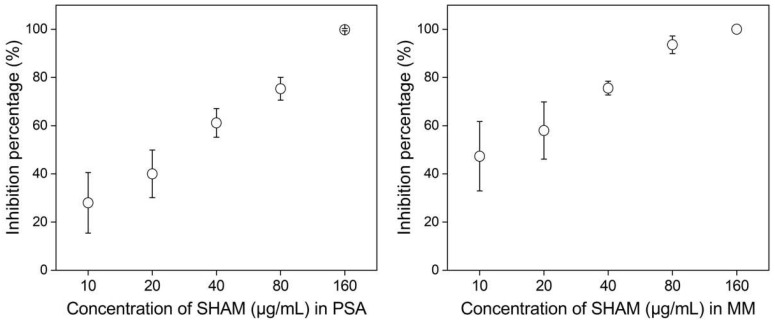
Dose response of mycelial growth of *Ustilaginoidea virens* to SHAM. Values are means of ten isolates; error bars represent standard errors of the mean. PSA, potato sucrose agar; MM, minimal medium.

**Figure 3 jof-08-01231-f003:**
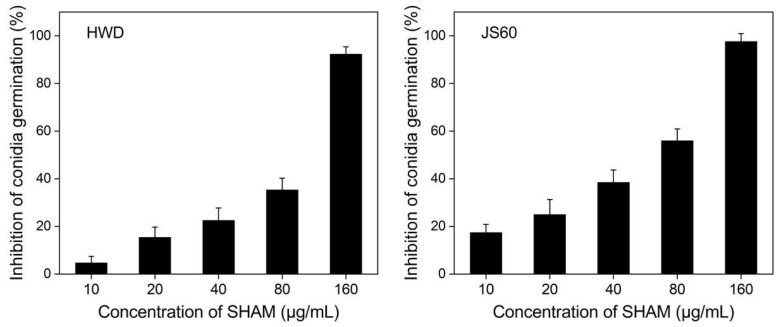
Inhibition of the conidial germination of *Ustilaginoidea virens* isolates HWD and JS60 by SHAM on water agar (WA). Error bars represent standard errors of the mean of independent three repetitions.

**Figure 4 jof-08-01231-f004:**
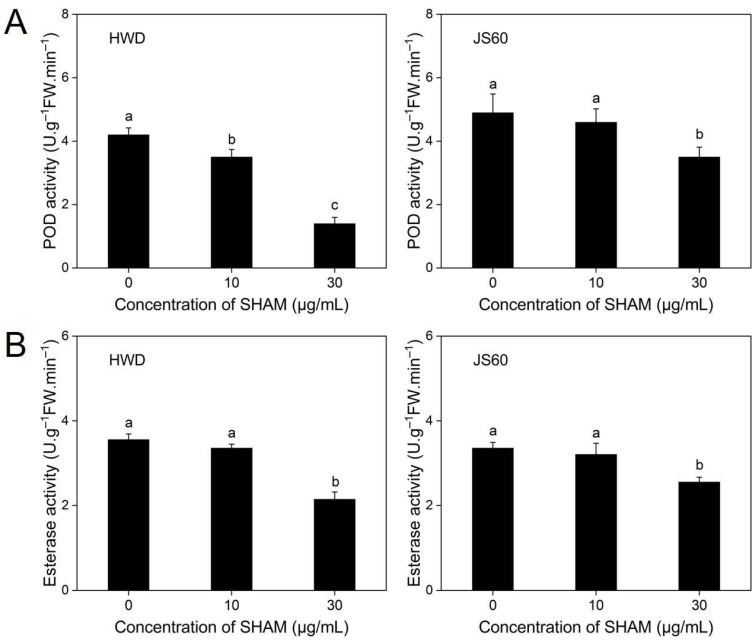
Effects of SHAM on activities of peroxidase (POD) (**A**) and esterase (**B**) of *Ustilaginoidea virens* isolates HWD and JS60. Error bars represent standard errors of the mean of independent three repetitions. Different letters are significantly different from each other (*p* < 0.05) based on one-way analysis of variance (ANOVA) with a least significant difference (LSD) test.

**Figure 5 jof-08-01231-f005:**
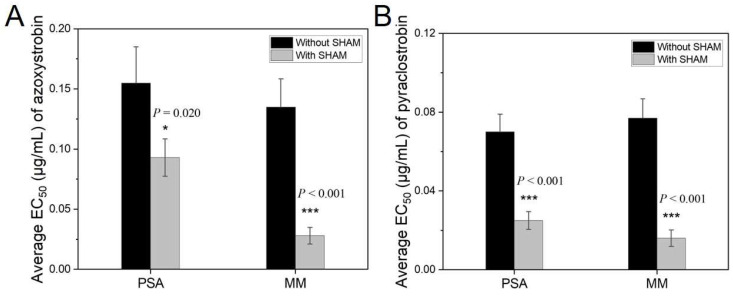
Effects of SHAM at 10 μg/mL on EC_50_ values of azoxystrobin (**A**) and pyraclostrobin (**B**) against mycelial growth of *U. virens* on PSA and MM. Values are means of ten isolates; error bars represent standard errors of the mean. Asterisk (* or ***) indicates that there is a significant difference (*p* < 0.05 or *p* < 0.001) between EC_50_ values with and without SHAM by paired *t* test. PSA, potato sucrose agar; MM, minimal medium.

**Table 1 jof-08-01231-t001:** Mycelial growth rate of *U. virens* and sensitivity to azoxystrobin and pyraclostrobin on potato sucrose agar (PSA) and minimal medium (MM).

Isolate	Mycelial Growth Rate (mm/d)			EC_50_ (μg/mL) of Azoxystrobin			EC_50_ (μg/mL) of Pyraclostrobin		
	PSA	MM	Significance ^z^	PSA	MM	Significance	PSA	MM	Significance
GL18	2.10	1.71	*	0.033	0.124	*	0.092	0.108	
GY20	2.13	2.02		0.089	0.120		0.084	0.038	*
HWD	2.02	1.83		0.251	0.198		0.046	0.081	
JR4	1.95	2.10		0.074	0.128		0.099	0.048	*
JS28	2.83	2.64		0.291	0.154	*	0.025	0.056	*
JS60	2.11	2.26		0.129	0.056	*	0.033	0.039	
YD2	1.85	1.43	*	0.279	0.141	*	0.118	0.112	
YZ10	1.36	1.29		0.126	0.119		0.063	0.088	
ZJ4	2.50	2.10	*	0.233	0.294		0.083	0.078	
ZJ26	2.24	2.26		0.051	0.019	*	0.058	0.123	*
Average	2.11	1.96	*p* = 0.430	0.156	0.135	*p* = 0.610	0.070	0.077	*p* = 0.615

^z^*T* test was used to compare the difference of mycelial growth rate and fungicide sensitivity between on PSA and MM. An asterisk (*) indicates that there is a significant difference (*p* < 0.05) for isolate between on PSA and MM by *t* test.

**Table 2 jof-08-01231-t002:** Effective concentrations for 50% inhibition (EC_50_) of SHAM against mycelial growth of *U. virens* on potato sucrose agar (PSA) and minimal medium (MM).

Isolate	EC_50_ (μg/mL) of SHAM		Fold Change	Significance ^z^
	PSA	MM		
GL18	40.18	17.97	2.24	*
GY20	28.83	18.53	1.56	*
HWD	27.24	13.83	1.97	*
JR4	15.30	7.55	2.03	*
JS28	13.39	5.92	2.26	*
JS60	29.29	17.91	1.64	*
YD2	30.69	12.52	2.45	*
YZ10	41.68	8.02	5.20	*
ZJ4	7.85	4.47	1.76	*
ZJ26	39.61	20.73	1.91	*
Average	27.41	12.75	2.30	*p* = 0.003

^z^*T* test was used to compare the differences in EC_50_ of SHAM between on PSA and on MM. An asterisk (*) indicates that there is a significant difference (*p* < 0.05) for isolate between on PSA and MM by *t* test.

**Table 3 jof-08-01231-t003:** Effects of SHAM at different concentrations on sensitivity of *U. virens* to azoxystrobin and pyraclostrobin.

Isolate	Concentration of SHAM (μg/mL)	Azoxystrobin				Pyraclostrobin			
		PSA		MM		PSA		MM	
		EC_50_ (μg/mL)	EC_90_ (μg/mL)	EC_50_ (μg/mL)	EC_90_ (μg/mL)	EC_50_ (μg/mL)	EC_90_ (μg/mL)	EC_50_ (μg/mL)	EC_90_ (μg/mL)
HWD	0	0.251	2.401	0.198	1.985	0.046	1.015	0.081	0.609
	10	0.148	2.236	0.039	0.543	0.031	0.970	0.019	0.202
	20	0.106	1.309	–	0.472	0.024	0.805	–	0.092
	40	–	0.925	–	0.097	–	0.346	–	0.041
	80	–	0.149	–	–	–	0.036	–	–
JS60	0	0.129	1.549	0.056	0.697	0.033	0.621	0.039	0.444
	10	0.117	1.537	0.020	0.516	0.018	0.477	0.012	0.113
	20	0.047	1.506	–	0.181	0.005	0.215	–	0.079
	40	–	1.068	–	0.043	–	0.044	–	0.011
	80	–	0.009	–	–	–	0.003	–	–

“–” indicated that the EC_50_ or EC_90_ value of fungicide cannot be calculated due to the inhibition of SHAM alone exceeded 50% or 90%. PSA, potato sucrose agar; MM, minimal medium.

## Data Availability

The data presented in this study are included in the article, further inquiries can be directed to the corresponding author.

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
