# Peer review of "The Toxicity of Salicylhydroxamic Acid and Its Effect on the Sensitivity of Ustilaginoidea virens to Azoxystrobin and Pyraclostrobin"

_jof, 2022, doi:10.3390/jof8111231_

Round 1

Reviewer 1 Report

The most important finding in this manuscript is that unlike other fungal pathogens, Ustilaginoida virens is sensitive to SHAM, therefore SHAM should not be included in the media for in vitro QoI sensitivity tests. This information is useful and should be beneficial for the readers.

The authors explained very well the background why the current study is important for the rice production in China. But they need to explain more about the experiments that they conducted (see below comments).

 Question:

Is there any reason why SHAM has such a strong effect on U. virens? Are AOX of U. virens different from those of other fungal pathogens?

Comments and suggestions:

L40-42: The fact that the majority of rice varieties are RFS-susceptible has nothing to do with the necessity of chemical control of RFS: the use of “Thus” doesn’t fit here. You need to add some phrases stating the importance of chemical fungicides to control RFS.

L62-65: (1) The AOX can be suppressed by plant secondary metabolites (2) SHAM in the media allows to evaluate more accurately the efficacy of QoI (3) “Thus?” the QoI can bemore consistent in planta efficacy? I don’t understand the logic here.

L78-79: Rephrase to remove “whether SHAM should and”.

L83: Spell out “PDA” and “MM”.

L89: Replace “per se” with “alone”.

L104: Remove “respectively”.

L110: Add reference for the MM recipe.

L112: EC50 values of what?

L115: One of the objectives of the study is to “select an appropriate medium for sensitivity test” but only two types of media, PSA and MM, are tested in this study. The authors should have included “PDA”, the most commonly used media to test QoI sensitivities in other fungal pathogens.

 L144-145: Why did you use only WA (water agar) for conidial germination assay? Why not MM and PSA?

L152: Spell out “POD”.

L152: POD and esterase are new information here. Explain the reason why you examined POD and esterase in Introduction section. Are they parts of AOX, and do you want to know if SHAM is affecting them?

L178: To test effects of SHAM at various concentrations,…

Table 1: t-test should be done for each isolate comparing the results between PSA and MM.

Figs.1 & 2. Spell out “PSA” and “MM” as the Tables and Figures should stand alone.

Table 2: t-test should be done for each isolate comparing the results between PSA and MM.

Fig. 3: Mention in the figure legend that this is done on WA media.

Tables 4 and 5. Some information is redundant with Table 1. You must remove the data redundancy.

L306: -> tests using

L308: Do we need a spectrometer for germ tube and germination assay?

L309-312: This information should be in Introduction to justify you used those media in the current study.

L338: -> is not always suitable..  

L369-370: This information should be in Introduction. This explains in part why you examined POD and esterase.

Author Response

Comments and Suggestions for Authors

The most important finding in this manuscript is that unlike other fungal pathogens, Ustilaginoida virens is sensitive to SHAM, therefore SHAM should not be included in the media for in vitro QoI sensitivity tests. This information is useful and should be beneficial for the readers.

The authors explained very well the background why the current study is important for the rice production in China. But they need to explain more about the experiments that they conducted (see below comments).

 Question:

Is there any reason why SHAM has such a strong effect on U. virens? Are AOX of U. virens different from those of other fungal pathogens?

Response: Thank you very much for your comments and suggestions to improve the manuscript. SHAM strongly inhibits U. virens, which we think may be due to two reasons. One is that AOX is essential for U. virens, and AOX of U. virens is very sensitive to SHAM. We will further study the functions of U. virens AOX in the future. Another is that SHAM can inhibit some essential enzymes which regulate mycelial growth and development of U. virens.

For another question, “Are AOX of U. virens different from those of other fungal pathogens?”. We compared the AOX protein (XP_042998005.1, 352 amino acids) of U. virens with that of other fungi. The homology comparison by BlastP showed that AOX of U. virens has up to 80.95% identity to that of Metarhizium anisopliae (KJK83722.1), 80.67% identity to that of Pochonia chlamydosporia (XP_018141496.1) and Metarhizium acridum (XP_007810939.1), 80.11% identity to that of Metarhizium robertsii (EXV01183.1), and 80.06% identity to that of Claviceps aff. purpurea (KAG6294413.1). The specific function of AOX in U. virens needs further study. We added this information in discussion.

Comments and suggestions:

L40-42: The fact that the majority of rice varieties are RFS-susceptible has nothing to do with the necessity of chemical control of RFS: the use of “Thus” doesn’t fit here. You need to add some phrases stating the importance of chemical fungicides to control RFS.

Response: Thank you. We rewrote the sentence as “Although planting resistant varieties is undoubtedly an economical and effective method to control RFS, the majority of rice varieties with good quality and high yields are RFS-susceptible. So far, fungicide-based chemical control is a significant measure to control RFS”.

L62-65: (1) The AOX can be suppressed by plant secondary metabolites (2) SHAM in the media allows to evaluate more accurately the efficacy of QoI (3) “Thus?” the QoI can be more consistent in planta efficacy? I don’t understand the logic here.

Response: To make it clear, we revised the sentence as “It has been assumed that including SHAM in artificial media can allow in vitro bioassay to be more consistency with in planta”.

L78-79: Rephrase to remove “whether SHAM should and”.

Response: We changed the sentence to “whether SHAM should be added remains unknown” based on your suggestion.

L83: Spell out “PDA” and “MM”.

Response: Revised it according to your suggestion.

L89: Replace “per se” with “alone”.

Response: Done.

L104: Remove “respectively”.

Response: We removed it.

L110: Add reference for the MM recipe.

Response: We added the reference.

L112: EC50 values of what?

Response: We revised it asEC50 values of azoxystrobin and pyraclostrobin against mycelial growth”.

L115: One of the objectives of the study is to “select an appropriate medium for sensitivity test” but only two types of media, PSA and MM, are tested in this study. The authors should have included “PDA”, the most commonly used media to test QoI sensitivities in other fungal pathogens.

Response: Thanks for your valuable suggestion. According to previous studies, PSA is a more suitable medium than PDA for mycelial growth of U. virens, and PSA is commonly used to test fungicide sensitivity in U. virens. Our preliminary experiment showed that the growth rate of U. virens on PDA is very close to that on PSA, and the sensitivity of U. virens to QoI fungicide on PDA is not significantly different from that on PSA. Considering that the components of these two media are relatively close, we only selected PSA for the whole study.

 L144-145: Why did you use only WA (water agar) for conidial germination assay? Why not MM and PSA?

Response: Firstly, WA (water agar) is the most commonly used medium for conidial germination assay. Secondly, WA has a single component, which has no effect on toxicity of SHAM.

L152: Spell out “POD”.

Response: Finished it.

L152: POD and esterase are new information here. Explain the reason why you examined POD and esterase in Introduction section. Are they parts of AOX, and do you want to know if SHAM is affecting them?

Response: In lines 74-76, there is relevant information in the introduction section. It has been reported that SHAM suppresses essential fungal enzymes, e.g., esterase, peroxidase, putative melanogenesis enzymes and laccase in other fungi. They are not parts of AOX, but plays an important role in fungal growth and resistance to oxidative stress. So we want to know if SHAM affects them in U. virens.

We revised the statements as “The versatile activities of SHAM against plant pathogens have been another issue of concern. AOX regulates mycelial growth, development, and resistance to oxidative stress. Apart from inhibiting AOX, SHAM also suppresses essential fungal enzymes, e.g., esterase, peroxidase, putative melanogenesis enzymes and laccase”.

L178: To test effects of SHAM at various concentrations, …

Response: When the concentration of SHAM was more than 30 µg/mL, the mycelium grows very little, especially on MM, which makes it difficult to collect enough mycelium. Therefore, the concentration of SHAM was set at 10 and 30 µg/mL, one of which is lower than EC50 of SHAM and the other is higher than EC50 of SHAM.

Table 1: t-test should be done for each isolate comparing the results between PSA and MM.

Response: We revised it.

Figs.1 & 2. Spell out “PSA” and “MM” as the Tables and Figures should stand alone.

Response: Yes. We finished it.

Table 2: t-test should be done for each isolate comparing the results between PSA and MM.

Response: We revised it.

Fig. 3: Mention in the figure legend that this is done on WA media.

Response: We added the information.

Tables 4 and 5. Some information is redundant with Table 1. You must remove the data redundancy.

Response: To avoid duplication of some data, we made Figure 5 to replace Table 4 and 5 in the manuscript, and moved Table 4 and 5 to supplementary information as Table S1 and S2.

L306: -> tests using

Response: Sorry for this mistake, we revised it.

L308: Do we need a spectrometer for germ tube and germination assay?

Response: Determination of conidial germination and growth by microtiter method need a spectrometer. So we revised the sentence as “Fungicide sensitivity tests using mycelial growth inhibition may not be as fast as testing conidial germination and growth or germination rates”.

L309-312: This information should be in Introduction to justify you used those media in the current study.

Response: Yes, we agree with you. So we added some information in the Introduction as “By mycelial growth inhibition, PSA is commonly used to determine the sensitivity of U. virens to fungicides. Nevertheless, the selection of different culture media can result in different result of sensitivity.”

But we want to keep this information in the Discussion section. Here we want to discuss that MM has not been used for U. virens before, and MM is an appropriate medium in our study. We added a sentence as “For U. virens, MM used as a culture medium has not been reported. In this study…”.

L338: -> is not always suitable…

Response:  Revised.

L369-370: This information should be in Introduction. This explains in part why you examined POD and esterase.

Response: As we mentioned above, there is relevant information in the introduction section.

Reviewer 2 Report

Manuscript ID jof-2025394

Manuscript title: Toxicity of salicylhydroxamic acid and its effect on the sensitivity of Ustilaginoidea virens to azoxystrobin and pyraclostrobin. Jiehui Song et al.

The authors had 3 main objectives: 1) select an appropriate medium to test sensitivity of U. virens to azoxystrobin and pyraclostrobin and SHAM toxicity. 2) investigate the toxic effect of SHAM per se on U. virens, and 3) assess the effect of SHAM on EC50 values of fungicides azoxystrobin and pyraclostrobin.

The manuscript is clear and well developed. The methodology used and the results achieved in this work are sufficient to be able to answer the questions asked by the authors.

The toxicity of salicylhydroxamic acid and its effect on the sensitivity of different pathogens to azoxystrobin and pyraclostrobin has been studied and reported for different pathogens, but this would be the first evidence for U. virens.

In my opinion, the toxic effect of SHAM is indisputable. In fact, I consider based on the large number of scientific works that describe it, that this test gives more headaches than solutions. I also consider that the effect of SHAM should only be tested on isolates resistant to QoI inhibitors. This would test whether resistance to QoI inhibitors is due to an AOX-dependent mechanism (for example, overexpression of enzymes involved in said mechanism). This aspect is not evaluated in the present work, as far as it can be seen the toxicity of SHAM is only evaluated in isolates susceptible to two QoI inhibitor fungicides. Here my question arises what happens in those isolates of U. virens resistant to these fungicides?. This aspect needs to be discussed and improved in the manuscript. If the authors have evidence that can complement my observation, the manuscript will be improved.

Author Response

Manuscript ID jof-2025394

Manuscript title: Toxicity of salicylhydroxamic acid and its effect on the sensitivity of Ustilaginoidea virens to azoxystrobin and pyraclostrobin. Jiehui Song et al.

The authors had 3 main objectives: 1) select an appropriate medium to test sensitivity of U. virens to azoxystrobin and pyraclostrobin and SHAM toxicity. 2) investigate the toxic effect of SHAM per se on U. virens, and 3) assess the effect of SHAM on EC50 values of fungicides azoxystrobin and pyraclostrobin.

The manuscript is clear and well developed. The methodology used and the results achieved in this work are sufficient to be able to answer the questions asked by the authors.

The toxicity of salicylhydroxamic acid and its effect on the sensitivity of different pathogens to azoxystrobin and pyraclostrobin has been studied and reported for different pathogens, but this would be the first evidence for U. virens.

In my opinion, the toxic effect of SHAM is indisputable. In fact, I consider based on the large number of scientific works that describe it, that this test gives more headaches than solutions. I also consider that the effect of SHAM should only be tested on isolates resistant to QoI inhibitors. This would test whether resistance to QoI inhibitors is due to an AOX-dependent mechanism (for example, overexpression of enzymes involved in said mechanism). This aspect is not evaluated in the present work, as far as it can be seen the toxicity of SHAM is only evaluated in isolates susceptible to two QoI inhibitor fungicides. Here my question arises what happens in those isolates of U. virens resistant to these fungicides? This aspect needs to be discussed and improved in the manuscript. If the authors have evidence that can complement my observation, the manuscript will be improved.

Response: Thank you very much for your kindly comments and suggestions. We agree with your opinion that the effect of SHAM should only be tested on isolates resistant to QoI fungicides. In our previous study, we tested the sensitivity of 179 U. virens isolates from different regions to two QoI fungicides, azoxystrobin and pyraclostrobin (Song et al. 2022, https://doi.org/10.1094/PDIS-12-21-2850-RE). Among them, no resistant isolates were found. Up to now, there are no reports on the resistance of U. virens to QoI fungicides. For this reason, it cannot be implemented at present that the effect of SHAM on those isolates of U. virens resistant to QoI fungicides. Your valuable opinion will be helpful to our future research which we will continuously monitor the emergence of QoI resistant isolates of U. virens in field or by fungicide-taming.